# CAN NEURON ACTIVATION BE PREDICTED? A NEW LENS FOR ANALYZING TRANSFORMER-BASED LARGE LANGUAGE MODELS

## ABSTRACT

Transformer-based large language models (LLMs) play a vital role in various NLP tasks, but the internal neurons are rather functioning in a black box style. In this work, we introduce the *Neuron Predictability Lens (NPL)*, an analytical framework that focuses on the way neurons work within feed-forward networks (FFNs). *NPL* is useful in understanding and analyzing transformer-based LLMs. Based on this framework, we conduct experiments on LLaMA-2 and GPT-J. Firstly, we show that neuron activations are predictable and for the first time we introduce the concept of *Neuron Predictability*. Secondly, we apply NPL to both global and local analysis. For global analysis, we investigate how FFNs contribute to model behaviors explicitly and implicitly with the aid of NPL. For local analysis, we explore the connection between neuron predictability and neuron interpretability. We examine various functional neurons under NPL and uncover the existence of "background neurons." With the findings mentioned above, we demonstrate the value of NPL as a novel analytical tool and shed light on its future application on model efficiency and/or effectiveness for improved language modeling.

## 1 INTRODUCTION

Large Language Models (LLMs) exhibit human-level proficiency in completing multiple natural language tasks (Vaswani et al., 2017; OpenAI, 2022; Touvron et al., 2023). However, these models are often regarded as "black boxes" since how their inner neuron function is mysterious (Bommasani et al., 2021). Insufficient understanding of LLMs hinders further optimization and responsible deployment of such powerful tools. Thus, paving the way towards a more transparent internal structure of LLMs becomes increasingly important. Efforts to understand and analyze LLMs range from global examinations of model behaviors to local dissections of specific modules (Luo & Specia, 2024). From a global view, researchers delve into comprehending the model's output and decision-making processes, e.g. detect how the activations in FFN contribute to the logits (Geva et al., 2021). In contrast, the local analysis seeks to unravel the mysteries of specific modules. For example, neuron interpretability research has dived into the relationship between individual neurons and specific linguistic tasks or functions (Dai et al., 2022a). Bridging these two perspectives, our work introduces a novel concept called *Neuron Predictability Lens*, which potentially encapsulates both the broader granularity and the finer granularity of LLM analysis with the discovery of *Neuron Predictability*. Figure 1 is an illustration of the concept.

*Neuron Predictability Lens (NPL)* is an analytical framework devised to provide a new perspective for understanding the behavior of transformer-based LLMs. NPL is performed through linear transformation, mapping FFN neurons across different layers. This method provides new insights, and renews the interpretability of vast concepts for transformer-based LLMs, such as logits contribution (i.e. the contribution of specific modules to the final logits, same hereafter) and neuron activation.

To make it clearer, we use *neuron activation* to denote the intermediate representation of the FFN module. We establish mappings between different layers and project activations in either a forward or a backward direction. We need to answer a natural research question (**RQ1**): *can neuron activation be predicted*? To answer this question, we train the neuron mappings across possible layer pairs on LLaMA-2 and GPT-J. Our experiments demonstrate that neuron activations are indeed pre-

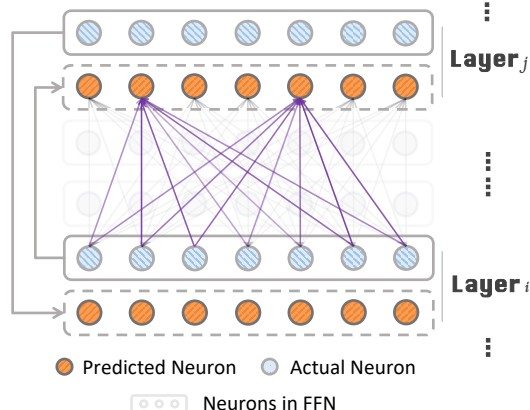

Figure 1: *Neuron Predictability*: The basis of *NPL*. The predicted version of neurons in layer $j$ (in orange) can be extracted from the actual neurons in layer $i$ (in blue), and vice versa.

dictably interconnected; the predictability persists even when transferring to data distribution away from the training data.

With the feasibility of neuron predictability established, we then raise the second research question (**RQ2**): *how to use NPL for model analysis?*

We utilize NPL to analyze LLMs in both global and local ways, unveiling findings in both branches: (1) In the global analysis, we substitute the predicted neuron activations for the actual ones and record the corresponding performance changes. Through this analysis, explicit and implicit contributions are investigated along with various substitution strategies. The main experiment reveals that shallow layers contribute to the final logits more implicitly while deep layers contribute more explicitly. The follow-up experiment delves deeper into the phenomenon and demonstrates that neurons with higher predictability are more crucial to the model performance. (2) Local analysis is conducted where we explore the relationship between neuron predictability and neuron interpretability. Through the lens of neuron predictability, we examine a variety of "functional" neurons pinpointed by prior research (Dai et al., 2022a), uncovering common characteristics among these functionally specialized neurons. From this analysis, we uncover "*background neurons*" – neurons that are vital to model performance, easy to predict, but do not exhibit explicit functional roles.

Overall, our contributions are as follows:

• **The NPL framework:** we propose and verify the effectiveness of *Neuron Predictability Lens* to analyze transformer-based LLMs;

• **Findings from the global analysis** with NPL: we find that shallow layers have more implicit logits contributions while deep ones have more explicit contributions; neurons with higher predictability contribute more to the final logits;

• **Findings from the local analysis** with NPL: our proposed method measures the predictability of functional neurons, and uncovers the existence of "background neurons."

## 2 NEURON PREDICTABILITY LENS

A major LLMs family is implemented based on transformer-based auto-regressive language models, which is our primary focus in this paper. Models are comprised of layers, and each layer contains two modules: a multi-head self-attention module (MHSA), and a FFN module. We define the outputs of MHSA and FFN of $l^{th}$ layer as $\mathbf{a}^l$ and $\mathbf{m}^l$ respectively. Then we have:

$$\mathbf{h}^{l+1} = \mathbf{h}^l + \mathbf{a}^l + \mathbf{m}^l, \tag{1}$$

where $\mathbf{h}^l$ denote the input vector of $l^{th}$ layer. Based on this equation, we can derive the formulation of the final representation:

$$\mathbf{h}^{final} = \mathbf{h}^1 + \sum_{l=1}^{L} \mathbf{a}^l + \sum_{l=1}^{L} \mathbf{m}^l. \tag{2}$$

In this work, we focus on the FFN module specifically, which has been proven to bear vast information (Suau et al., 2020; Geva et al., 2021; 2022; Dai et al., 2022a; Wang et al., 2022; Luo & Specia, 2024; Gurnee et al., 2024). The inner structure of FFN comprises two full-connection feed-forward layers with the activation function sandwiched between them. Formally:

$$\text{FFN}(\mathbf{x}) = \mathbf{W}^O \cdot \sigma\left(\mathbf{W}^I \cdot \mathbf{x}\right), \tag{3}$$

where $\sigma$ is the activation function, and $\mathbf{W}^I \in \mathbb{R}^{d \times d_{\text{ffn}}}$ and $\mathbf{W}^O \in \mathbb{R}^{d_{\text{ffn}} \times d}$ are learnable weight matrices. $d$ is the hidden size and $d_{\text{ffn}}$ is the intermediate dimension of FFN. For simplicity, the bias terms of linear layers are ignored.

**Neurons in FFN**  NPL is proposed based on the Neurons in FFN. To elaborate the neurons, we rewrite Equation 3 as:

$$\text{FFN}(\mathbf{x}) = \sum_{i=1}^{d_{\text{ffn}}} [\mathbf{g}]_i \mathbf{W}_{:,i}^O, \tag{4}$$

$$\mathbf{g} = \sigma(\mathbf{W}^I \cdot \mathbf{x}).$$

Just like the previous studies (Dai et al., 2022a; Wang et al., 2022; Zhang et al., 2023), neurons are defined here as the column vectors $\mathbf{W}_{:,i}^O$. We denote $\mathbf{g}$ as the activation vector, indicating the activation of neurons. The $i^{th}$ element of $\mathbf{g}$ is the activation of the $i^{th}$ neuron.

The *Neuron Predictability* indicates a mapping between neurons in different FFN modules. Given two layers $i$ and $j$, we establish projection $M_{i \to j} : \mathbb{R}^{d_{\text{ffn}}} \to \mathbb{R}^{d_{\text{ffn}}}$ which projects from the activation vector $\mathbf{g}_i$ of layer $i$ to the activation vector $\mathbf{g}^j$ of layer $j$. From this projection, we could get $\tilde{\mathbf{g}}^j = M_{i \to j}(\mathbf{g}^i)$, where $\tilde{\mathbf{g}}^j$ is a predicted item of real $\mathbf{g}^j$. NPL measures how well $\tilde{\mathbf{g}}^j$ fits $\mathbf{g}^j$. We use two metrics to evaluate the prediction, the L2 distance and the Pearson Correlation (Pearson, 1895). The prediction mapping is implemented by a linear transformation and is optimized by minimizing the mean square error (MSE). Below are the corresponding equations.

$$M_{i \to j}(\mathbf{g}^i) := \mathbf{W}_{M_{i \to j}} \cdot \mathbf{g}^i \tag{5}$$

$$\mathbf{W}_{M_{i \to j}} = \arg\min_{\mathbf{W}} \mathbb{E} ||\mathbf{W} \cdot \mathbf{g}^i - \mathbf{g}^j||_2 \tag{6}$$

## 3 PRELIMINARY ANALYSIS: PREDICTABILITY OF NEURON ACTIVATIONS

In this section, we implement NPL in real settings to answer **RQ1**. The results prove the existence of neuron predictability in tested models.

### 3.1 EXPERIMENTAL SETUP

We establish mapping $M_{i \to j}$ across every other layer on LLaMA-2-7b (Touvron et al., 2023) and GPT-J-6b (Wang & Komatsuzaki, 2021) ($\forall i, j \in \{2k \mid 2k < L, k \in \mathbb{N}\}$; $L$ is the number of layers). Not all layers are utilized due to constraints by computational resources. Here, $i$ could be either smaller than, larger than, or equal to $j$.

We use the training set of WikiText2 (Merity et al., 2016) to train the mappings. Since a quick and consistent convergence emerges while training, we sample a subset (about $10^7$ tokens) instead of using the entire dataset in the real process. We employ the Adagrad optimizer (Duchi et al., 2011) and set the initial learning rate as 0.01. The training is completed for a single epoch with a batch size of $10^4$ tokens. All experiments are conducted on one A100.

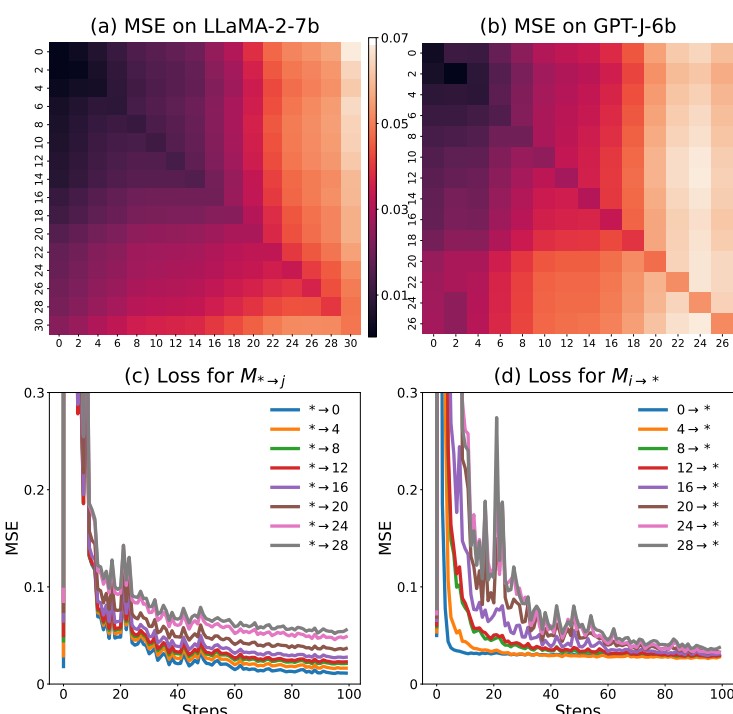

Figure 2: (a, b) Performance evaluation of learned neuron mappings for LLaMA-2 and GPT-J using the MSE score. Appendix A.1 provides more interpretations of these MSE results; (c, d) Averaged training curve on LLaMA-2. $M_{*\to j}$ denotes mapping from any layer to layer $j$, and $M_{i\to *}$ denotes mapping layer $i$ to any layer. The L2 distance of random activation is 0.0971 (with a std of 5.154e-6 over five runs) and the cosine similarity is 0.

## 3.2 Results and analysis

Figure 2 (a, b) is the visualization of NPL implementation. The figures illustrated the layer-wise neuron predictability on LLaMA-2-7b and GPT-J-6b. The predictability is measured by L2 distance. In the results, the overall L2 distances are around or less than 0.05, and the largest L2 distance is no more than 0.07. This decent result shows that neurons demonstrate a predicting relation between layers, and the phenomenon exists in both models. Figure 2 (c,d) visualize the training process of the NPL mappings. The MSE losses decrease more than $10\times$ when converge, which indicates the effectiveness of the learned NPL mapping.

The predictability varies among layers. Shallow layers tend to yield better predictability than deeper ones regardless of the projecting direction. Similar results are shown in the averaged training loss in Figure 2 (c, d), where shallow layers converge quicker and better in both projecting directions.

Furthermore, we calculate the average L2 distance for three different cases: 0.037 for shallow-to-deep prediction ($i < j$), 0.024 for deep-to-shallow prediction ($i > j$), and 0.020 for self-prediction ($i = j$). These results indicate that deep-to-shallow prediction is more accurate than the reverse, with self-prediction yielding the best performance. This means deeper layer FFN activations encapsulate information from shallower layers, which accounts for the greater ease of predicting shallower layer outputs from deeper FFN activations. (In addition to L2 distance, we also calculate the cosine similarity as the evaluation metric, which gives us similar results. See in App. B.1)

We conduct a follow-up experiment on cross-domain generalization. Results in Table 1 show that the NPL framework performs well in different tasks. We provide more interpretations of the table below in App.A.2. We also investigate the neuron predictability of different models and context lengths, which you can see in App. C.

Table 1: Averaged performance of Chunks 1-4 on WikiText-2, Alpaca Taori et al. (2023), and XSum Narayan et al. (2018).

| Mapping | Substitution | WikiText-2 | Alpaca | XSum |
|---------|-------------|-----------|--------|------|
| *Random* | Complete | $> 200$ | $> 200$ | $> 200$ |
|          | Partial | 55.69 | 13.49 | 12.37 |
| *NPL* | Complete | 42.67 | 12.44 | 12.16 |
|       | Partial | 38.53 | 9.76 | 8.21 |
| Original | | 33.35 | 8.63 | 6.23 |

## 4 GLOBAL ANALYSIS: ANALYZING THE LOGITS CONTRIBUTION OF PREDICTED NEURONS

This is our first step to answer **RQ2**. Through NPL, we evaluate how the predicted activations affect the model performance, which both provides a global LLM analysis and validates the effectiveness of NPL. Specifically, we substitute the actual neuron activations with those predicted by the NPL Mapping. Given a mapping $M_{i \to j}$ where the activation of layer $i$ serves as the stimulus for predicting the response in layer $j$, we substitute the authentic activations in layer $j$ with the predicted ones.

### 4.1 EXPERIMENTAL SETUP

Recalling Equation 2, due to the existence of residual connection, the final representation of the model $\mathbf{h}^{final}$ can be viewed as a summation of the outputs from the FFN and MHSA modules of each layer. This final representation is normalized and projected to the "logits" over vocabulary via the language modeling head. We refer to the FFN output $\mathbf{m}^l$ as the **explicit contribution** from the FFN$^l$ to the logits as $\mathbf{m}^l$ is explicitly added to the final output $\mathbf{h}^{final}$. There is also an **implicit contribution** from FFN$^l$, as deeper layer representations are computed based on the outputs of those shallower layers. Therefore, $\mathbf{m}^l$ also contributes to $\mathbf{h}^{final}$ implicitly by involving the computation of all its subsequent layers.

In this section, we conduct substitution experiments to study how the predicted neuron activations affect the explicit and implicit contributions. Figure 3 is the visualization of the two settings. As forward propagation proceeds from shallow layers to deep ones, we only consider the mapping $M_{i \to j}$, i.e. when $i < j$ (if not specified, we set $i = j - 1$ in the rest of the paper).

We split all layers into four chunks and enumerate them from shallow to deep. In each trial, we substitute neuron activations of one chunk of layers. For each setting, the following three types of mappings are compared with *NPL Mapping*.

• ***Random Mapping*** substitutes actual activations with activations obtained through a randomized mapping. In the random mapping, we run the evaluation 5 times and compute the average.

• ***Zero Mapping*** zero-outs actual activations.

• ***Identical Mapping*** substitutes actual activations with activations from its previous layer.

### 4.2 RESULTS AND ANALYSIS

Table 2 presents the results extracted in various substitution settings in LLaMA-2 and GPT-J. The *NPL Mapping* exerts the most negligible impact on the logits, corroborating that neuron predictability indeed captures information intrinsically linked to the model's capabilities. In contrast, the *Random Mapping* and *Zero Mapping* either introduce meaningless noise or remove the activations within certain chunks, both resulting in a substantial perturbation of the logits.

There is a strong correlation between the depth of substituted layers and the resultant effect. Substituting activations within the two middle chunks causes a relatively minor impact on the final logits, whereas substitution at either the bottom or top chunks introduces a more pronounced effect.

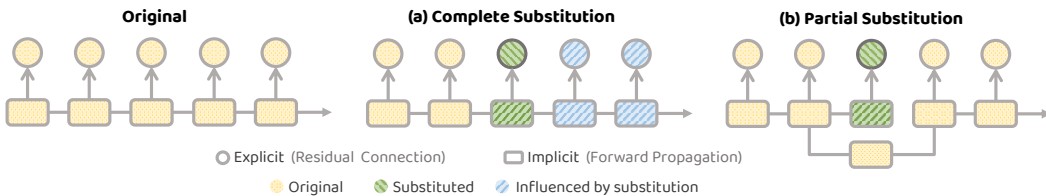

Figure 3: An illustration of the substitution settings in the global analysis: (a) *Complete Substitution* where both explicit and implicit contributions are substituted, (b) *Partial Substitution* where only implicit contribution is substituted.

Table 2: Perplexities of various settings. The perplexity of LLAMA-2 and GPT-J without substitution is 33.08 and 26.58, respectively. We calculate the sentence-level perplexity upon sentences with *varied length*. We also provide full results on Alpaca and XSum in App. B.3.

| Settings | | LLaMA-2 | | | | GPT-J | | | |
|---|---|---|---|---|---|---|---|---|---|
| Mapping | Substitution | Chunk 1 | Chunk 2 | Chunk 3 | Chunk 4 | Chunk 1 | Chunk 2 | Chunk 3 | Chunk 4 |
| *Random* | Complete | > 1000 | 47.54 | 54.49 | 55.82 | > 1000 | 403.02 | 45.23 | 580.48 |
| | Partial | 33.98 | 33.37 | 42.52 | 112.89 | 25.09 | 27.77 | 53.09 | 282.60 |
| *Zero* | Complete | > 1000 | 43.56 | 50.07 | 55.14 | > 1000 | 383.30 | 43.88 | 557.31 |
| | Partial | 33.33 | 33.40 | 42.34 | 109.47 | 24.98 | 27.38 | 49.32 | 286.14 |
| *Identical* | Complete | > 1000 | 58.96 | 65.19 | 62.35 | > 1000 | 94.99 | 41.20 | 246.34 |
| | Partial | 34.83 | 33.56 | 43.59 | 114.07 | 24.21 | 27.48 | 58.72 | 231.28 |
| *NPL* | Complete | 47.23 | 37.78 | 38.51 | 47.17 | 247.76 | 58.73 | 34.92 | 46.82 |
| | Partial | 33.41 | 33.54 | 37.49 | 49.66 | 23.64 | 24.60 | 32.01 | 50.30 |

Additionally, our findings indicate that this correlative relationship manifests differently when assessing explicit versus implicit contributions.

Here is a bulleted list of our findings:

- **Only FFN in deep layers (Chunk 4) exhibit a significant explicit contribution to the logits.** Conversely, substituting the activations in the shallow layers, particularly layers in Chunk 1, demonstrate an almost negligible explicit contribution to the logits regardless of the substitution setting.

- The trend is reversed for implicit contributions. **FFN in shallow layers (Chunk 1) contribute more implicitly than those in deep layers (Chunk 4)**. Since the shallow layers play foundational roles and influence all the subsequent computations, this phenomenon is explainable. Thus, if these layers are compromised, the ability of the model would be severely impaired. On implicit contribution, *NPL Mapping* shows an evident advantage over other substitution strategies, again suggesting that NPL captures anticipated meaningful semantic information to some extent.

- Another intriguing finding is that in Chunk 4, *complete substitution* outperforms *partial substitution* in all mappings for LLaMA-2 and in *NPL Mapping* for GPT-J. This counterintuitive phenomenon suggests that **in deep layers, the presence of a "fake" explicit contribution appears to elicit a negative effect on the actual implicit contribution**.

### 4.3 FINER-GRAINED NEURON SUBSTITUTION

In this part, we further investigate the performance of neurons with different predictability. This time, for each layer in one chunk, more/less predictable neurons are substituted by different strategies. Two distinct metrics are utilized to guide our selection: the L2 distance and the Pearson correlation. Figure 4 shows the results of the experiment.

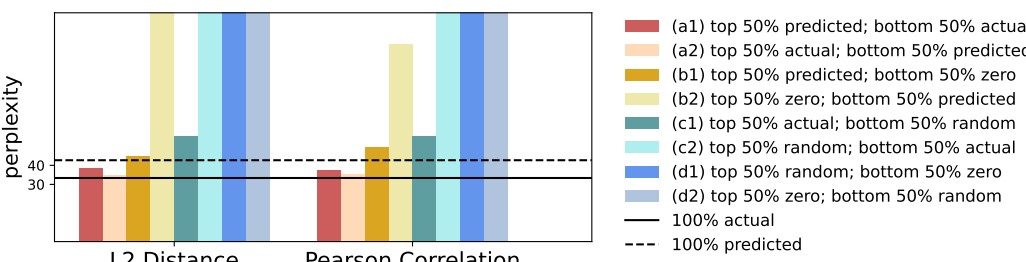

Figure 4: Comparison of perplexities for different neuron substitution strategies. We categorize all neurons into two groups based on two metrics and then implement substitution strategies. For instance, *(b1) top 50% predicted; bottom 50% zero* indicates that the 50% more predictable neurons under the chosen metric are replaced with *predicted* activations, while the rest are set to *zero*. Here we report the averaged results of all chunks.

**Results**   Figure 4 shows both the effectiveness of the prediction and the relationship between neuron predictability and model performance. The substitution of either more or less predictable neurons extracts a similar performance to the actual as shown in (a1) and (a2). Comparing (b1) with (b2), we find that preserving the information of 50% more predictable neurons is sufficient for maintaining acceptable performance, even if the remaining neurons are masked. Also, there is a strong relationship between predictability and perplexity as shown in (b) and (c). Neurons being easier to predict tend to be more important to the model performance.

**Insights**   The experiment leads to several findings: (1) The predicted neuron activations are effective in retaining model performances. (2) Neurons with higher predictability are more important for the model performance. (3) The correlation between neuron predictability and the importance of the neuron implies that *NPL Mapping* is not random but rather related to neurons' intrinsic properties.

### 4.4   INTERIM SUMMARY

With NPL, the above global analysis delves into LLMs' inner structures by detecting corresponding contributions to the model's logits. Apart from the analysis itself, this section validates the effectiveness of the NPL framework as the neuron predictability indeed captures information relevant to model capability instead of learning irrelevant features.

## 5   LOCAL ANALYSIS: ANALYZING THE PREDICTABILITY OF FUNCTIONAL NEURONS

This section demonstrates how the NPL framework could adapt to the local LLMs analysis and steps further to answer **RQ2**. We classify neurons according to their specialties and detect the predictability of functional neurons. Following the previous works, we use the term "functional neuron" to denote neurons whose activation patterns correlate to a specific function, such as token identification, position encoding, knowledge storing, etc. (Gurnee et al., 2024; Voita et al., 2023; Dai et al., 2022b). We conduct further analysis on functional neurons and examine their characteristics under NPL. To this end, we first locate functional neurons, and then evaluate their predictability. We follow the procedure of Gurnee et al. (2024) to locate these neurons. For a given neuron $i$ in layer $l$, we compute:

$$\mu_{\mathcal{P}}^{l,i} = 1 - \frac{(1-\beta)\sigma^2([\mathbf{g}^l]_i|\mathcal{P}(\mathbf{x})) + \beta\sigma^2([\mathbf{g}^l]_i|\neg\mathcal{P}(\mathbf{x}))}{\sigma^2([\mathbf{g}^l]_i)}, \tag{7}$$

where $\mathcal{P}$ represents the property function that determines whether the input token $\mathbf{x}$ exhibits the specific functionality, and $\beta$ is the proportion of tokens that possess this functionality. The resulting $\mu_{\mathcal{P}}^{l,i}$ serves as the importance score of neuron $i$ in layer $l$ concerning functionality $\mathcal{P}$. Neurons with

the functionality are those extracting higher $\mu_{\mathcal{P}}^{l,i}$. Afterward, we compute the mean difficulty score $S_{\mathcal{P}}^l$ of predicting those neurons:

$$S_{\mathcal{P}}^l = \frac{1}{|N_{\mathcal{P}}^l|} \sum_{i \in N_{\mathcal{P}}^l} s^{l,i}, \tag{8}$$

where $N_{\mathcal{P}}^l = \{i | \mu_{\mathcal{P}}^{l,i} > \theta_{\mathcal{P}}\}$ is the subset of filtered neurons and $s^{l,i}$ is the difficulty score of predicting neuron $i$ in layer $l$. This difficulty score is measured by L2 distance. A lower $S_{\mathcal{P}}^l$ indicates an easier prediction of neurons with property $\mathcal{P}$, which means they have higher predictability. For comparison, we also compute the difficulty score on all evaluation data and on a random subset of tokens as shown in Fig. 5 (a). The series of experiments shows the predictability of the functional neurons. The presence of the specific functionality is considered a sufficient condition for high activation. In each following section, we examine one specific kind of functional neuron. All results in this section are with complete substitution.

### 5.1 N-GRAM-SENSITIVE NEURONS $\mathcal{P}_{n-gram}$

Some neurons are found to activate exclusively when specified $n$-grams are present in the input, as a result, they are named as "$n$-gram detecting" neurons (Voita et al., 2023).

We examine $n$-grams with $n$ ranging from 1 to 3, conduct a comprehensive analysis of all $n$-grams presented within the test corpus, filter out meaningless ones, and select the 1,000 most frequent ones for each $n$ for further investigation. As shown in Fig. 5 (b), there is a clear distinction between the predictability of $n$-gram sensitive neurons and the random baseline across most of the layers. Different $n$ extract similar difficulty scores. This means $n$-gram sensitive neurons are difficult to predict regardless of the choice of $n$. This result shows the consistent feature of predictability of the $n$-gram sensitive neurons.

### 5.2 DIFFICULTY-SENSITIVE NEURONS $\mathcal{P}_{loss}$

Neuron prediction could be easily associated with token prediction, which leads our investigation toward difficulty-sensitive neurons. We found that the activations of certain neurons are correlated with the performance of the causal language modeling objective. Tokens that are hard to predict, manifesting in high cross-entropy loss (denoted as *hard tokens*), tend to activate specific neurons. Similarly, there are the *easy tokens* which are activated in response to tokens that are easy to predict.

We filter the tokens based on their cross-entropy loss, and then get the hard and easy tokens. As depicted in Fig. 5 (c), difficulty-sensitive neurons exhibit significantly higher scores than the random baseline for all the conditions, which means they are harder to predict. Furthermore, it is observable that the neurons corresponding to hard tokens exhibits greater difficulty when being predicted.

The identification of difficulty-sensitive neurons is intriguing. These functional neurons are harder to predict, and those responding to hard tokens are even harder to predict. As hard tokens represent greater challenges for the model, their information flow within the model would be complicated, making their prediction more difficult. We also examined the hardest and easiest 10000 tokens in the same experiment (See App. B.4).

### 5.3 POSITION-SENSITIVE NEURONS $\mathcal{P}_{pos}$

Another branch of neurons is those associated with positional information, which activates in response to the position rather than the token or its context. Inspired by Voita et al. (2023), we hypothesize that positional neurons can work in teams and collectively respond to various positional patterns. Based on this hypothesis, we explored two types of positional pattern: (1) the *arbitrary* pattern includes a randomly-sampled subset of all positions; (2) the *successive* pattern includes a fixed-length span of consecutive positions. We clip the maximum input length to 1024 and examine positions ranging from 1 to 1024. As illustrated in Figure 5 (d), only the scores of successive patterns exhibit significant deviations from the random baseline. This means there are neurons with the special function of successive-position detection, and their activations are hard to predict. Results of another positional pattern are shown in Appendix B.5.

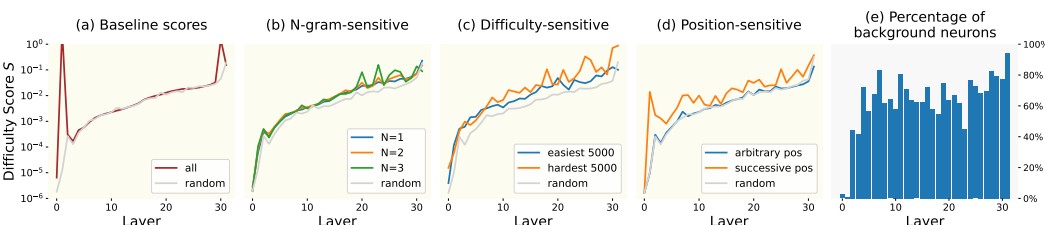

Figure 5: Results of functional neuron experiments in LLaMA-2. (a-d) shows the difficulty score for predicting the neurons. All values pass the significant test with p < 0.001. Please note that the scale of the y-axis is logarithmic. The numerical results are shown in App. B.2; (e) shows the percentage of background neurons across different layers.

## 5.4 "BACKGROUND" NEURONS

In the above examination, all neurons associated with specific functionalities exhibit high difficulty scores, indicating that they are hard to predict. Conversely, we are also interested in those more predictable neurons. To this end, we set the random baseline as a threshold of the difficulty score and get those neurons with higher predictability. As depicted in Figure 5 (e), a substantial proportion of neurons (ranging from 40% to 80%) fall into this category. Appendix B.6 provides perplexities after replacing the background neurons and the layer-wise statistics of background neurons. This suggests that a majority of the neurons within FFNs are relatively easy to predict. The precise function of these neurons is challenging to define, but their critical importance is evident based on the results after masking them out.

The masking experiment is conducted in the same setting described in Section 4 and the results are averaged over four chunks. The perplexity after masking those background neurons is 170.80. Compared with the original perplexity of 33.08, the performance decays significantly after masking the background neurons. Masking the same amount of random neurons causes less severe degradation and extracts a perplexity of 55.53. The mysterious nature of background neurons shows that a considerable proportion of neurons contribute to model behavior while "working in the dark." This prompts us to rethink how we credit the success of the model's performance.

## 5.5 INTERIM SUMMARY

In this section, a variety of functional neurons are examined through NPL. Functional neurons tend to have a consistent feature of lower predictability. Besides, a large number of neurons have high predictability, and do not have defined functional roles, but are vital for model performance. Thus, we name them as the "background neurons." We also examined outlier neurons in Appendix B.7.

## 6 DISCUSSIONS AND IMPLICATIONS

The above analysis reveals NPL as an effective analytical tool for LLMs. Here, we discuss the following applications and implications.

First, NPL would help inference acceleration by short-cutting transformers. Previous research has investigated inference acceleration by establishing linear shortcuts across transformer blocks (Din et al., 2023). NPL bears a resemblance to these efforts by predicting the neuron activations in FFN without significant performance drop, suggesting the potential of NPL as a promising avenue for bypassing the complicated computations of vanilla transformers.

Second, NPL can uncover causal relationships between neuron activations across different layers. For instance, if a later-layer neuron's activation is precisely predictable from early-layer neuron activations, we can infer causal links between these neurons. By integrating NPL with existing circuit discovery techniques, we can enhance the mechanism analysis of LLMs.

Third, NPL encourages us to rethink the role of FFN. Some researchers posit FFN functions as key-value memories (Geva et al., 2021), while others suggest it projects hidden representations onto a distribution over the output vocabulary, thus amplifying the predicted probability of some words while diminishing that of others (Geva et al., 2022; Belrose et al., 2023; Katz & Belinkov, 2023). Our investigation reveals these arguments to be incomplete. FFNs at various depths play diverse roles, and even within the same layer, individual neurons exhibit varied behaviors.

## 7    RELATED WORK

Though the proposal of NPL is initial, its implementations are built on previous research. As analyzing transformers has attracted much attention in recent years, researchers have delved into this intricate structure with multiple methods. Following Luo & Specia (2024), we roughly categorize transformer analysis into two streams: local analysis, which delves into the intricacies of individual transformer components, and global analysis, which seeks a holistic understanding of the behaviors and capabilities of the model.

Among local analysis, Dai et al. (2022a) shed light on the storage of knowledge within model parameters by identifying specific "knowledge neurons." Similarly, Voita et al. (2023) uncover a range of functional neurons characterized by regular activation patterns. They target individual neurons and experiment on their functionalities. Global analysis encompasses a variety of approaches, including probing techniques (Rogers et al., 2020; Petroni et al., 2019; Li et al., 2023), mechanistic interpretability (Elhage et al., 2021; Wang et al., 2023), and more. Among these, the "Vocabulary lens," which projects weights and activations onto the vocabulary space, is a trending analytical tool (Geva et al., 2021). This lens allows researchers to explore how different modules and inputs contribute to model performance(Belrose et al., 2023; Ram et al., 2023; Geva et al., 2023). Another direction is to analyze transformers through simple mappings between modules. For example, Dar et al. (2023) learn to project parameters into a shared embedding space, while Din et al. (2023) explore linear shortcuts between layers, which bears conceptual relevance to our approach. Different from previous studies, our introduction of the neuron predictability lens encompasses both the local and global facets of transformer analysis.

## 8    CONCLUSION

In this work, we discover the predictability of neuron activations and present the Neuron Predictability Lens (NPL) as a powerful analytical framework for examining transformer-based LLMs.

Through extensive experiments, the predictability of neuron activations has been demonstrated and significant insights have been uncovered into the contributions of different layers to the final logits. The global analysis highlights the distinct roles of shallow and deep layers, while the local analysis in this paper sheds light on the existence and importance of "background neurons" in LLMs.

The contribution of our NPL framework in analyzing LLMs is unique. Moving beyond traditional approaches, we offer a new perspective for this research line. The NPL framework has the potential to uncover the predictable relations within the LLM models, providing a new lens for LLM studies.

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

Below is the additional information we provide for better understanding. The appendix includes three parts: Additional Interpretations, Additional Results, and Additional Experiments.

# A    ADDITIONAL INTERPRETATIONS

## A.1    EVALUATION OF PREDICTABILITY USING MSE

To rethink the rationality of MSE evaluation, we propose and test the following hypotheses toward a sounder evaluation of predictability using MSE:

- **Hypothesis A: Finer Granularity** Our current version of MSE calculates the averaging mean-squared error of all neurons in the layer, which is rather coarse-grained. In Sec. 4.3, we have shown that neurons in the same layer have diverse predictability. Therefore, a proper metric should capture such characteristics.
- **Hypothesis B: Magnitude-invariance** The metric should be magnitude-invariant, especially in a scenario where the neuron activations are small.

Then we evaluate the effectiveness of the above hypotheses. We calculate the neuron-wise Relative Squared Error (RSE) for comparison.

We conduct substitution experiments to see how these metrics fit the model performance. Specifically, we filter top/bottom neurons with metrics and then substitute them with zero/predicted activation. The perplexity scores are shown in Table 3.

Table 3: Perplexity scores of the substitution experiments for MSE interpretation.

| | Top 1000 | | | Bottom 1000 | | |
|---|---|---|---|---|---|---|
| Metrics | Mean | PPL. (Zero) | PPL. (Predicted) | Mean | PPL.(Zero) | PPL. (Predicted) |
| Tensor-Wise MSE | — | — | — | — | — | — |
| Neuron-Wise MSE | 0.0073 | 171.7 | 33.88 | 0.0016 | >1000 | 36.25 |
| Neuron-Wise RSE | 0.6719 | >1000 | 41.36 | 1.1094 | 210.4 | 35.28 |

From the experiment, we can infer that (a) **hypothesis A is correct** because there exists a significant difference between the performance of high/low MSE/RSE neurons; (b) **hypothesis B is wrong** because neuron-wise MSE aligns better with the performance than RSE.

Then, *why do magnitude-invariant metrics fail to evaluate predictability?*

Magnitude-invariant metrics normalize the difference between predicted values and the actual ones so that large and small values can have a "fair" comparison. For example, for two variables $x$, $y$ we have $x_{actual} = [100, 120]$, $x_{predict} = [90, 125]$, $y_{actual} = [0.1, 0.12]$, $y_{predict} = [0.09, 0.125]$. The two variables have the same RSE. Behind these metrics, there lies a presupposed assumption: large values can tolerate larger deviations while small values bear less.

The predictability is not suitable to be measured with magnitude-invariant metrics because the neuron activation does not follow the assumption. There is no evident correlation between the magnitude of neuron activation and robustness against deviations. Thus, a small RSE does not guarantee a better approximation in terms of performance. This means our evaluation with MSE is effective when detecting the predictability.

## A.2    CROSS-DOMAIN GENERALIZATION OF NPL

To demonstrate that *NPL Mapping* does not just imitate the distribution of the training data, we evaluate its cross-domain generalization ability. The experimental setting is the same as Section 4. As shown in Table 1, while trained on Wikitext, *NPL Mapping* successfully generalizes to other data distributions by outperforming *Random Mapping* and closely approximating the performance of real activations. These results suggest that NPL is not limited to the specificities of the training data but rather captures broader, more universal patterns that are applicable even in contexts that diverge from the original training domain or language.

## A.3 LIMITATIONS OF OUR WORK

Since NPL is a newly proposed analytical framework, more applications are to be explored. Our work is an initial attempt to analyze transformers with NPL, and even at this early stage, we have already uncovered interesting insights. Due to space limitations, some experimental results are not fully elaborated. We include them in the appendix. Moreover, we use linear mapping to implement the NPL framework, while other kinds of mappings could also be explored, though this would likely incur additional computational overhead. Future research may explore other mappings to further leverage the potential of NPL.

## B ADDITIONAL RESULTS

### B.1 RESULT OF COSINE DISTANCE AS A METRIC

We add cosine distance as the additional metric, which is consistent with L2 distance, indicating the effectiveness of neuron predictability.

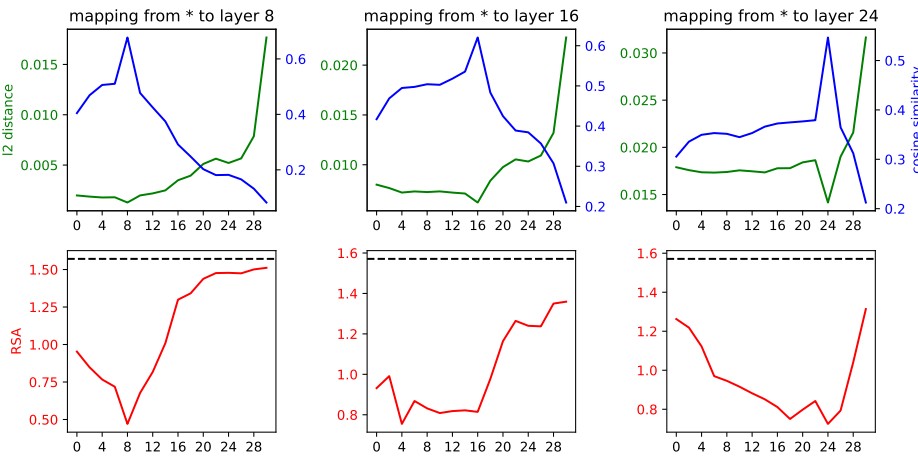

Figure 6: L2 distance, cosine similarity, and representational similarity analysis (RSA) of layer prediction in LLaMA-2. For RSA analysis, we use Centered Kernel Alignment (CKA) for evaluation. The black dashed lines in RSA plots represent the RSA distance of the random noise.

### B.2 NUMERICAL RESULTS ON FUNCTIONAL NEURONS

Table B.6 includes the numerical results in different experimental settings. Note that all values pass the significance test with $p < 0.001$.

We calculate the following results: (a) the averaged result of N-gram sensitive neurons, (b) the results on easy/hard tokens of the Difficulty Sensitive neurons, (c) the results of the successive pattern for the Position Sensitive neurons, and (d) the results of the random setting.

Table 4: Numerical results on functional neurons

|  | N-gram (average) | Difficulty (easy) | Difficulty (hard) | Position (successive) | Random |
|---|---|---|---|---|---|
| Averaged Predictability | 0.024 | 0.025 | 0.095 | 0.037 | 0.012 |

### B.3 ADDITIONAL RESULTS OF THE SUBSTITUTION EXPERIMENT IN SECTION 4

Full results for the substitution experiment on Alpaca and XSum are shown in Table 5.

Table 5: Full results for the substitution experiment on Alpaca and XSum.

| Mapping | Substitution | Chunk 1 | Chunk 2 | Chunk 3 | Chunk 4 |
|---------|-------------|---------|---------|---------|---------|
| *Alpaca* | | | | | |
| *Random* | Complete | > 1000 | 11.98 | 11.44 | 12.12 |
| | Partial | 9.16 | 8.78 | 10.38 | 25.62 |
| *NPL* | Complete | 17.68 | 10.37 | 10.27 | 11.44 |
| | Partial | 8.67 | 8.83 | 9.63 | 11.91 |
| *XSum* | | | | | |
| *Random* | Complete | > 1000 | 10.38 | 11.56 | 13.47 |
| | Partial | 6.94 | 6.68 | 9.08 | 26.78 |
| *NPL* | Complete | 19.57 | 8.45 | 9.57 | 11.06 |
| | Partial | 6.63 | 6.67 | 8.30 | 11.25 |

## B.4 ADDITIONAL DETAILS ON DIFFICULTY-SENSITIVE NEURONS

We extract results from four settings in the experiment. As shown in Figure 7 (a), neurons responding to: 1) the hardest 5000 tokens, 2) the easiest 5000 tokens, 3) the hardest 10000 tokens, and 4) the easiest 10000 tokens are in detection. The first two conditions are elaborated in Sec. 5.2. All difficulty scores are higher than the random baseline. When we scale up target tokens, the difficulty score fluctuates across layers. The predictability associated with neurons corresponding to hard tokens exhibits greater fluctuations. This suggests LLMs possess a form of self-awareness regarding the confidence in predicting the next tokens. By probing its internal representations, we can uncover such "mental states" of LLMs without external signals.

Figure 7: Full results on difficulty-sensitive neurons and position-sensitive neurons.

## B.5 ADDITIONAL DETAILS ON POSITION-SENSITIVE NEURONS

In the real settings, we explored three types of positional patterns: (1) the *arbitrary* pattern includes a randomly-sampled subset of all positions; (2) the *successive* pattern includes a fixed-length span of consecutive positions; (3) the *oscillatory* pattern includes selected positions at regular intervals. The oscillatory pattern shows a slight difference from the random baseline, which means they do not present a lower predictability in our experiment. Results are shown in Fig. 7 (b).

To be mentioned, Voita et al. (2023) has uncovered the oscillatory positional neurons. However, their definition of "FFN neurons" is different from the "neurons" in our work. Thus, though there is no difference between the predictability of oscillatory positional neurons and the random baseline, this could not demonstrate the predictability feature of the former discovered neurons.

### B.6 ADDITIONAL DETAILS ON BACKGROUND NEURONS

We replace the activation of background neurons with zero/mean/noise. The conclusion is that all three ablations hurt the performance badly. We represent the results as follows:

| Original | Zero | Mean | Random |
|----------|--------|--------|--------|
| 33.08 | 170.80 | 145.21 | 153.34 |

From these results, we can conclude that background neurons are not just for adjusting norms. There is rich information hiding behind them, which requires further investigation.

Figure 8 shows that background neurons are easier to predict (with lower L2 distance) and have a large quantity (painted with yellow).

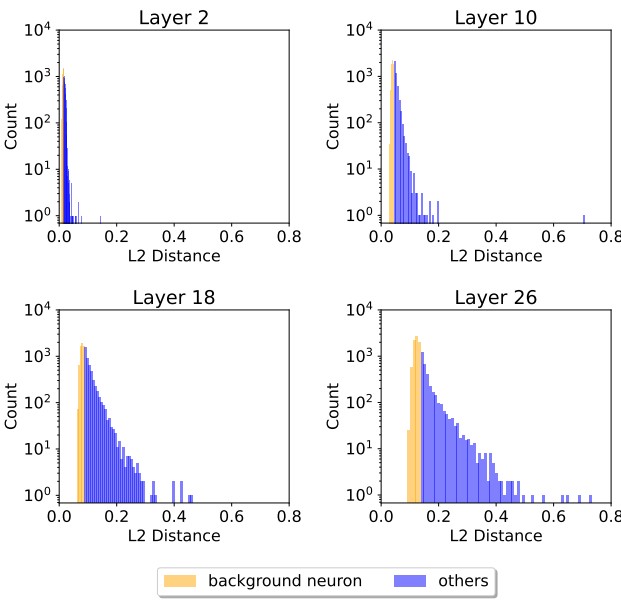

Figure 8: Statistics of background neurons in layer 2, 10, 18, 28 in LLaMA-2-7b.

### B.7 ADDITIONAL DETAILS ON OUTLIER NEURONS

Outlier phenomenon has been observed across various LLMs (Puccetti et al., 2022). This phenomenon refers to the persistent emergence of extreme values within the models' activations and weights which, though comprising less than 0.1% of the values, can exceed the magnitude of other values by several hundredfold and are thus termed "outliers".

For LLaMA-2-7b, we find the 7890-th neuron of layer 2 (shorted as L2.7890) to be an outlier. We observe that the occurrence of outliers is associated with meaningless tokens, such as <SOS>, <UNK>. As for neuron predictability, outlier neurons are extremely hard to predict.

## C ADDITIONAL EXPERIMENTS

### C.1 CROSS-MODEL NEURON PREDICTABILITY

Neuron mapping can be established not only within a single but also across different models. To validate this, we conduct experiments applying NPL between the LLaMA-2-7b and LLaMA-2-13b models. Figure 9 shows that the neuron mapping across models is learnable. Our observations reveal a strong correlation between the layers of the two models, with the most effective mappings establishing when layers of similar depth are used to predict each other. Additionally, based on the

L2 distance metric, we have noted that shallower layers tend to be more predictable than their deeper counterparts, a similar phenomenon observed in single-model experiments.

## C.2 CONTEXT LENGTH AFFECTS NEURON PREDICTABILITY

We investigate scenarios where tokens are exposed only to a constrained segment of the preceding context. To achieve this, we employ a context window, denoted by $w$, to limit the range of context accessible to each token. Subsequently, we train multiple NPL mappings for various $w$ values and visualize the differences. As shown in Figure 10, a larger $w$ extends the context scope and also results in increased predictability for neurons in shallower layers, while simultaneously decreasing predictability in deeper layers. We hypothesize that an extended context provides the NPL with more comprehensive information, aiding in the accurate prediction of neuron activations in shallow layers. Contrastingly, the semantics in deeper layers may become too complex to be captured by the NPL.

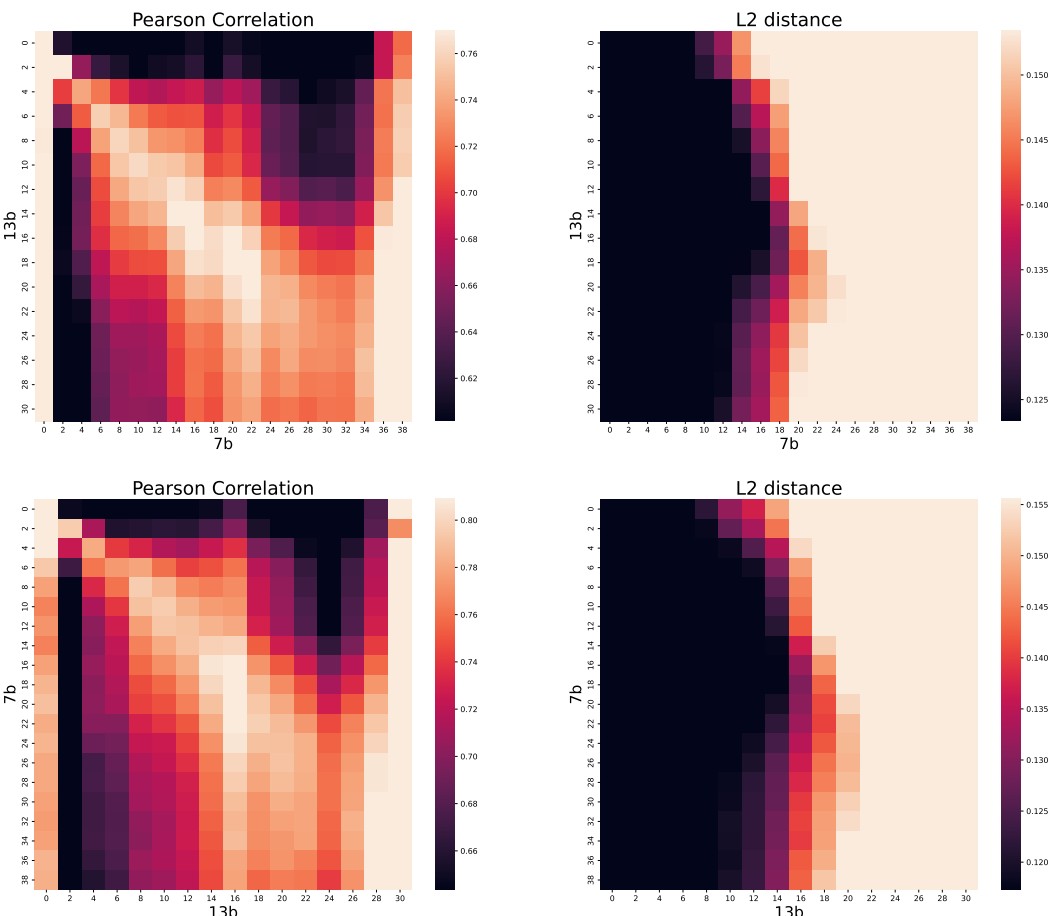

Figure 9: NPL between LLaMA-2-7b and LLaMA-2-13b.

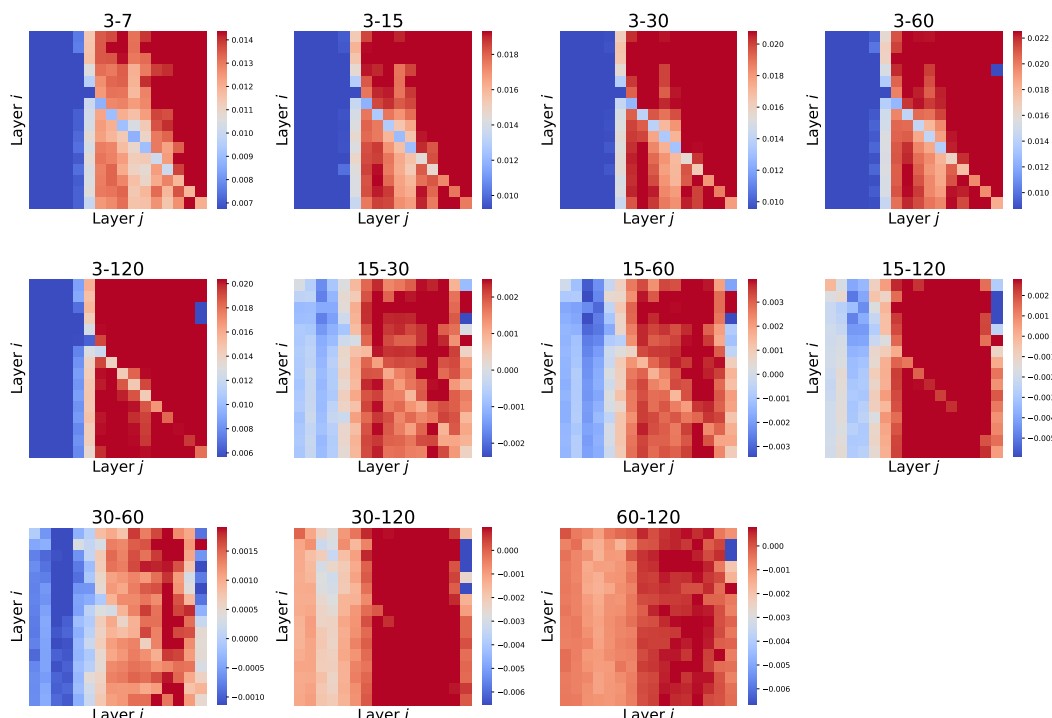

Figure 10: L2-distance difference of NPL mappings on LLaMA-2-7b under the settings of different window sizes $w$. The window sizes selected for this analysis include $w \in \{3, 7, 15, 30, 60, 120\}$.

