# OpenReview forum: "Can Neuron Activation be Predicted? A New Lens for Analyzing Transformer-based LLM"
_ICLR.cc/2025/Conference — ICLR 2025 Conference Withdrawn Submission_

### Official Review · Reviewer_d8Xc · 2024-11-01

**Soundness:** 2
**Presentation:** 2
**Contribution:** 2
**Rating:** 5
**Confidence:** 2

**Summary:**

The paper introduces the Neuron Predictability Lens (NPL), an analytical framework designed to analyze the internal workings of neurons within feed-forward networks (FFNs) in transformer-based large language models (LLMs). The authors conduct experiments on LLaMA-2 and GPT-J, demonstrating that neuron activations are predictable and uncovering various insights through global and local analyses using NPL. The paper contributes to the understanding of transformer-based LLMs and suggests potential applications of NPL in model efficiency and effectiveness.

**Strengths:**

1. The paper introduces a new analytical framework, NPL, which focuses on the predictability of neuron activations in transformer-based LLMs. This is a unique perspective that has not been explored extensively in previous research.
2. The authors conduct both global and local analyses using NPL, providing a comprehensive understanding of the internal workings of transformer-based LLMs. The findings from these analyses are insightful and contribute to the existing knowledge on transformer-based models.

**Weaknesses:**

1. The experiments are conducted only on LLaMA-2 and GPT-J, which may limit the generalizability of the findings. It would be beneficial to see experiments on a wider range of transformer-based models to validate the effectiveness of NPL.
2. The method lacks novelty, which essentially is a linear regression between neuron activations of different layers.

**Questions:**

How does the NPL change over different chunks of input texts? Also the input length has an effect on NPL. More analysis could be included

---

### Official Review · Reviewer_f4GH · 2024-11-02

**Soundness:** 2
**Presentation:** 1
**Contribution:** 3
**Rating:** 3
**Confidence:** 4

**Summary:**

This paper proposes a new framework for analyzing Transformer-based model’s neuron activations toward performance and functionalities. Based on the predictability of neuron concept, this paper conducts comprehensive analyses in global- and local-levels. The results of the analyses implicate correlation between predictability of neurons and performance & functionalities.

**Strengths:**

Simple idea of modeling neuron predictability: The idea is simple, which is learning linear mapping between a state (neurons) to another state at different feed-forward network (FFN) layers.

Interesting analysis results: Some results of analyses sound new and interesting. For example, easy-to-predict neurons are more important for performance than other hard-to-predict neurons.

Potential to contribute future works of large language model field:
As described in section 6, these analyses could contribute to other works for better computational efficiency if some concerns (in Weaknesses) are addressed well.

**Weaknesses:**

Major:
- Experiment results: The PPL of original LLAMA-2 and GPT-J (33.08 and 26.58 in Table 2), on wikitext2 are too high regarding their large number of parameters. For example, GPT-J (6B) is known to be better than GPT-2 (1.5B) (check https://huggingface.co/EleutherAI/gpt-j-6b) , but GPT-2 (1.5B) achieved 18.34 PPL on wikitext2. I guess the trained large language models did not converge well, or maybe the simplified forward propagation (no consideration of layer normalization) is the key reason.
- Lack of explanations: I believe that this paper could be improved by explaining more clearly some parts as follows:
: In section 3, to select which dimension of state to substitute, this paper used L2 distance and Pearson correlation. However, it is unclear what are the two objects used to compute distance (or correlation).
: Equations 7, 8 are introduced without explanations of some notations, such as ‘g’ with square brackets, sigma square (maybe variance?), operation ‘|’ with ‘ㄱ’ shape symbol, and ‘theta_P’.
: The difficulty score, ‘s^{l,i}’ in Equation 8, is indexed with only i-th neuron and l-th layer. However, because this means difficulty for predictability, I guess it needs additional index to indicate ‘from which layer’s neuron is used to predict’.
: For the baseline of Figure 5’s experiments, this paper utilized the difficulty score of a random subset of tokens. Firstly, it is hard to understand what is the random subset of tokens. Also, the difficulty score requires a property function (P) to filter out neurons. This is not explained in the paragraph within lines 384~391. Also, it is not clear why this could be the global baseline for n-gram sensitive, difficulty (loss) sensitive, position sensitive analyses?
: Each property functions, such as n-gram and position sensitivity, need clearer definitions. Also, it was hard to understand what ‘arbitrary’ and ‘successive’ positions with only text.

Minor:
- Less intuitive naming for two main perspectives: For example, ‘global view’ (relationship between neuron activation and model performance) and ‘local view’ (relationship between neuron activation and linguistic functions) are less intuitive.
- Location of the paragraph (line 213~215) looks improper in section 3.2, because it contains the contents of Section 4 (Table 1, ‘Complete’, ‘Partial’ substitutions and ‘Original‘ models). It is hard to follow this paragraph.
- Need clearer explanations: For example, about the findings in section 4.2, the authors might want to clearly mention which type of substitution (complete or partial) they are talking about.

**Questions:**

- In Figure 2 (a, b), the MSE losses of diagonal terms indicate the performances of linear mappings learned to predict the neuron activation (hidden state of FFN) of a layer given the same hidden state of that layer. This sounds like a trivial task and can be perfectly solved by identity weight matrix. However, in results, the diagonal MSE performances are not 0.0, and some of other off-diagonal linear mappings achieved better MSE performances. Could you explain what I am missing here?

- Why the predictor is a linear model? Can we use multi-layer perceptrons?

---

### Official Review · Reviewer_FVRc · 2024-11-05

**Soundness:** 2
**Presentation:** 2
**Contribution:** 2
**Rating:** 5
**Confidence:** 3

**Summary:**

This paper introduces the Neuron Predictability Lens (NPL), a framework designed to analyze the predictability of neuron activations in Transformer-based large language models (LLMs). The authors apply this framework to examine neuron activations in LLaMA-2 and GPT-J, demonstrating predictability across layers. The paper presents both a global analysis, showing contributions of specific layers to model outputs, and a local analysis, uncovering distinctions in neuron functionality and identifying "background neurons" that are vital yet functionally ambiguous.

**Strengths:**

- The NPL framework provides a fresh perspective on interpretability for Transformer-based LLMs, particularly by focusing on predictability as an avenue to understand neuron roles in model behavior.
- The study includes an extensive evaluation across multiple model configurations and datasets, offering a robust examination of neuron predictability.
- The findings contribute meaningfully to the field, especially the discovery of "background neurons" that, while predictable, do not exhibit clear functional roles. This insight is likely to spark further investigation into latent neuron functionality in LLMs.

**Weaknesses:**

- While the NPL framework provides a novel lens, its practical applications, especially beyond interpretability, are not clearly outlined. Integrating this framework into tasks that directly improve model performance or efficiency would enhance the relevance of the work.
- The experiments could benefit from a wider range of models or additional comparisons with other interpretability frameworks. This would clarify the uniqueness and generalizability of the NPL approach across diverse architectures.
- Although the analysis of "background neurons" is insightful, the study of functional neurons is a well-trodden path in neuron interpretability. Incorporating additional layers of functional analysis could further validate the utility of NPL.

**Questions:**

- The distinction between global and local analysis with NPL is well-conceived, providing a structured approach to dissect both overarching and fine-grained model behaviors. However, more connection between the two analyses would enhance the coherence of the findings.
- It remains unclear if predictability could translate into actionable insights for model optimization, such as inference acceleration or memory reduction.
- Elaborate on potential real-world applications, especially in model efficiency and performance, where predictability could serve as a metric for optimization.
- To contextualize NPL, compare its findings and utility directly against existing interpretability tools, such as neuron ablation studies or vocabulary lenses.
- Introduce a more granular categorization or additional characterization of functional neurons, potentially linking neuron predictability with specific tasks or contextual dependencies.

---

### Note · Authors · 2024-12-14

I have read and agree with the venue's withdrawal policy on behalf of myself and my co-authors.